# Intra- and Inter-Rater Reproducibility of Measures of Physical Performance in Patients with COPD

**DOI:** 10.3390/jcm14196755

**Published:** 2025-09-24

**Authors:** Christina Nielsen, Nina Godtfredsen, Stig Molsted, Charlotte Ulrik, Henrik Hansen

**Affiliations:** 1Respiratory Research Unit Hvidovre and Department of Respiratory Medicine, Copenhagen University Hospital—Hvidovre, Kettegaard Allé 30, 2650 Hvidovre, Denmark; nina.skavlan.godtfredsen@regionh.dk (N.G.); charlotte.suppli.ulrik@regionh.dk (C.U.); henrik.hansen.09@regionh.dk (H.H.); 2Institute of Clinical Medicine, University of Copenhagen, Blegdamsvej 3B, 2200 Copenhagen, Denmark; stig.moelsted@regionh.dk; 3Department of Clinical Research, Copenhagen University Hospital—North Zealand, Dyrehavevej 29, 3400 Hilleroed, Denmark; 4Department of Rehabilitation Sciences and Physiotherapy, University of Antwerp, Prinsstraat 13, 2000 Antwerpen, Belgium

**Keywords:** COPD, performance test, outcome assessment, 1MSTS, handgrip strength, SPPB, reproducibility

## Abstract

**Background**: Assessments of functional capacity require considerable time and space and are thus generally not suitable for home-based assessments. Reproducibility measures from tests evaluating functional capacity suitable for home-based assessment are warranted. The aim of this study was to investigate the intra- and inter-rater reproducibility of home-based assessments of the one-minute sit-to-stand test (1MSTS), handgrip strength (HGS), and short physical performance battery (SPPB) in patients with moderate to very severe chronic obstructive pulmonary disease (COPD). **Methods**: Fifty patients were recruited from an ongoing RCT study prior to the start of the intervention. All patients performed two 1MSTS attempts with 30 min intervals, three HGS attempts, and one SPPB attempt. The procedure was repeated after 7–10 days by a blinded assessor. **Results**: Fifty patients (29 females; mean (SD): age 71.3 (±7.7) years, FEV1% predicted 37.4 (±14.1), and COPD assessment test score 19.4 (±6.5) were assessed. The 1MSTS intra-rater reliability (intraclass correlation coefficient; ICC_1.1_) was 0.95 (lower limit 95% CI: 0.91) and classified as excellent; agreement (standard error of measurement, SEM) was 1.7 repetitions. The inter-rater reliability ICC_1.1_ for 1MSTS was 0.91 (lower limit 95% CI: 0.84) and SEM 2.5 repetitions; for HGS it was ICC_1.1_ of 0.84 (lower limit 95% CI: 0.74) and SEM 4.2 kg; for SPPB it was ICC_1.1_ of 0.86 (lower limit 95% CI: 0.77) and SEM 0.9 points, respectively. **Conclusions**: The reproducibility of home-based assessment with 1MSTS, HGS, and SPPB in patients with moderate to very severe COPD showed good to excellent intra- and inter-rater reliability and acceptable agreement below the established minimal important change. These findings support the utility and feasibility of these tests as reliable outcome measures in clinical practice and their applicability in home-based settings enabling functional assessments of patients with transportation challenges and mobility limitations.

## 1. Introduction

Pulmonary rehabilitation (PR) is strongly recommended for patients with chronic obstructive pulmonary disease (COPD) [1]. In rehabilitation it is important to have reliable performance tests to measure changes of physical and functional capacity [2]. The psychometric strength of the tests should be assessed in disease populations and must demonstrate reproducibility to accurately distinguish between measurement error and true changes across repeated measures. Population-specific reproducibility parameters are important to ensure accurate interpretations in research and clinical practice [3,4,5].

The 6 min walk test (6MWT) is used in research and clinical practice to assess functional and walking capacity [6,7]. However, the 6MWT requires considerable space and time and may thus reduce repeated routine use for assessments in a home-based setting and in primary care. Alternative performance tests that require less space and equipment have been proposed to increase the accessibility for patients who are unable to complete first-choice standard walking tests [8,9]. The use of practical, simple assessments that reflect physical performance in patients with COPD are warranted, especially in settings where traditional field tests may not be feasible [8,10].

The 1 min sit-to-stand test (1MSTS) has been proposed as a suitable alternative to the 6MWT to estimate functional capacity in patients with COPD [8], and there is a strong correlation between 1MSTS and 6MWT [11,12]. The two tests elicit similar cardiovascular responses on heart rate and dyspnea score [8,13], making the 1MSTS suitable for assessing functional capacity [9]. Evidence on intra-rater and inter-rater reproducibility of 1MSTS in COPD remains limited. One validation study reported an intra-class correlation coefficient (ICC) of 0.99 (95% CI: 0.97–1.00) with a minimal important difference (MID) of three repetitions [14]. A second study found an intra-rater ICC of 0.90 (95% CI: 0.80–0.95) and no systematic bias was found [8]. Research on inter-rater reliability is scarce and no studies have investigated reproducibility in a home setting.

Handgrip strength (HGS) is widely used to evaluate general muscle weakness and physical frailty among patients with COPD [15]. Reduced HGS is also associated with all-cause mortality [16] and reduced health-related quality of life [17]. However, the reproducibility of HGS in patients with COPD has been tested in only two studies; one study has reported an intra-rater reliability among patients with COPD and found an ICC_2.1_ of 0.99, a standard error of measurement (SEM) of 0.59 kilogram force (Kgf), and a clinically acceptable minimal detectable change (MDC) of 1.64 Kgf (95% limits of agreement (LOA) (−2.5 to 2.1 Kgf)) [18]. Medina-Mirapeix et al. reported an inter-rater reliability in a hospital setting with an ICC of 0.97 (95% CI: 0.93–0.98) and SEM of 1.3 kg (kg) [19].

The short physical performance battery (SPPB) is designed to measure physical performance and evaluate balance, lower limb strength, and gait speed over a short distance. The SPPB provides valuable insight into functional limitations, physical frailty, and identification of risk or disability [20,21]. Physical frailty assessment has demonstrated significant utility in identifying frail individuals, evaluating changes following PR, and predicting all-cause mortality [20]. Reproducibility values for the SPPB in COPD populations are limited. One study assessed interobserver reliability in patients with COPD in a hospital setting [19] and found an ICC of 0.92 (95% CI: 0.62–0.91) and a SEM of 0.55 points for the total score; further tests are needed to evaluate its value in a home setting.

Whilst performance tests are typically conducted in a controlled clinical setting, the tests can be performed in a home setting with challenges and as well as advantages: the home setting is a less controlled environment; however, it provides a familiar place for the patient, that may reduce anxiety associated with clinical visits. As reported above, more research on reproducibility of physical performance tests performed in a home-based setting is warranted.

The aim of this study was to evaluate the intra- and inter-rater reliability as well as the limits of agreement for the 1MSTS, HGS, and the SPPB in patients with moderate to very severe COPD in a home-based setting.

## 2. Materials and Methods

### 2.1. Study Design

This intra- and inter-rater reproducibility study was conducted as a part of the multicenter randomized controlled trial (RCT) “Supervised pulmonary tele-rehabilitation and individualized home-based pulmonary rehabilitation for patients with COPD, unable to participate in center-based programs. The protocol for a multicenter randomized controlled trial—the REPORT study” [22]. Trial registration at ClinicalTrials.gov, identifier NCT05664945. We followed the Guidelines for Reporting Reliability and Agreement Studies (GRAAS) [5].

### 2.2. Participants

Participants eligible for the REPORT study were identified and recruited from respiratory departments of seven university hospitals in the Capital Region of Denmark (Amager, Bispebjerg, Bornholm, Frederikssund, Gentofte, Hvidovre, and Hilleroed), two municipalities (Copenhagen and Frederiksberg), and via a project site on Facebook. All patients signed a written informed consent form. All patients recruited for the REPORT study were consecutively invited to participate in the intra-and inter-rater reproducibility study. Data collection commenced on 18 January 2023 and continued until a full sample size of 50 patients was reached on 26 November 2024 (Figure 1). A sample size of 50 patients was chosen based on the recommendations of Consensus-based Standards for the selection of health Measurement INstruments (COSMIN) [23], as this number balances practical feasibility with the statistical precision needed to obtain reliable estimates of measurement properties.

Inclusion criteria were the following: indication for PR according to national guidelines; unable to access and participate in the conventional out-patient hospital- or community-based PR when offered during routine consultation; no participation in PR the past 24 months; COPD defined as a postbronchodilator FEV_1_/FVC < 0.70 and a FEV_1_ < 80% pred (moderate to very severe airflow limitation); GOLD group B or E; able to stand up from a chair and walk 10 m independently (with or without a walking aid); and lift both arms to a horizontal level with a minimum of 1 kg dumbbells in each hand. Exclusion criteria as described in the study protocol [22].

### 2.3. Study Settings and Raters

A patient was given the choice of performing the tests at the hospital or at home, to accommodate their individual preferences. When testing was in the hospital, transport was provided by taxi to/from the home to avoid fatigue prior to the test.

All tests were performed by nine experienced physiotherapists (PTs) who assisted as blinded raters from four different hospitals. All raters had experience in conducting performance tests and collecting data; median years of experience as PTs were 13 years (<10 years [n = 3], 10–20 years [n = 4], and over 20 years [n = 2]); and areas of medical expertise included heart and lung diseases. All raters completed an assessment calibration course before the tests were performed.

One rater conducted the assessments on test day one (T1), and another rater carried out the assessment on test day two (T2). The second rater was blinded to the results from T1. Prior to T2, patients were informed by the investigator not to reveal their randomization nor results from T1 to ensure blinding.

### 2.4. Test Procedures

Prior to assessments, the patients were instructed to take their medication as prescribed and avoid vigorous physical activity three hours before. All assessments during T1 and T2 followed the same procedures, time frame, and setting. The interval from T1 to T2 was 7–10 days and disease stability over this period was assumed. Walking aids (e.g., a rollator or crutch) were allowed to be used during the gait test in SPPB. Patients on long-term oxygen therapy were allowed to use them during tests. Oxygen consumption (L/min) was recorded. The complete test battery of the original study consisted of physical tests and questionnaires, and total time for completion was approximately 1.5 h. See Table 1 for full overview of the assessment order.

1MSTS was performed using an armless chair with a height of 44–46 cm, and the patient’s arms crossed and placed at the chest. The test was demonstrated by the rater. The patient was asked to perform 1–2 practice attempts before performing the test (one min rest). The patient was instructed to perform as many sit-to-stand repetitions as possible. A notification was given 15 s before the end of the test; no other communication was provided during the test. The patient was allowed to pause during the test, but with no stopping of time. If the patient was not able to perform sit-to-stand with armed crossed, the scoring was recorded as 0 repetitions. A 30 min pause between the first and second trial was mandatory in this study, with the higher repetitions of the two trials recorded in number used for the reproducibility analysis. Heart rate (HR), arterial oxygen saturation measured by pulse oximetry (SpO_2_), and perceived dyspnea (Borg cr-10) were assessed before and after each 1MSTS trial.

HGS tests were performed on both hands, starting with the dominant hand in a standardized position, seated in a straight back chair with the feet flat on the floor, shoulder at 0° abduction, flexion and rotation, the elbow flexed at 90°, and forearm and wrist rested in a neutral position between supination and pronation [18]. The test was repeated three times on both hands with 15 s rest between trials to avoid fatigue. The highest value (measured in kg) obtained from the dominant hand was used in the reproducibility analysis. A Jamar Hydraulic Hand Dynamometer, model SH5001, Saehan Corporation, Masan, Korea, was used.

SPPB was performed in the following sequence: (a) standing balance test (Guralnik), (b) a 3-m gait speed (3MGS), and (c) a five-times sit-to-stand test (5STS test). The balance assessment included three tests: side-by-side stance for 10 s, semi-tandem stance for 10 s, and tandem stance for 10 s. To advance to the subsequent stance, the patient was required to maintain the initial stance for 10 s. Upon successful completion, the patient progressed to the second stance, and if this was also achieved, advancement to the final stance followed. In the 3MGS test the patient was instructed to walk at normal gait pace; the time used was recorded and the test was performed twice (use of walking aid recorded). The fastest trial was used for the reproducibility analysis. The 5STS was performed with arms crossed at the chest with no use of the arms for support. The patient was instructed to perform the five repetitions as fast as possible. The test was performed twice with 2 min apart to avoid fatigue.

### 2.5. Other Variables

The following demographic and descriptive variables were collected: sex, age, height, weight, body mass index, smoking status (former, current, and ex-smoker), years with COPD, GOLD group, MRC, FEV_1_/FVC, FEV_1_, prescribed respiratory medication, LTOT, Charlson co-morbidity index, BODES index, walking aid, dyspnea, and exercise capacity. Respiratory symptoms were assessed by the COPD assessment test (CAT).

### 2.6. Statistical Analyses

Descriptive statistics are presented as means with standard deviation (SD) for continuous data and as medians with range for ordinal data and data not normally distributed. Data distribution was inspected via histograms and Q–Q plots and verified by Shapiro–Wilk test to test for approximately normal distribution. Paired *t*-test and Wilcoxon signed rank test were used to compare systematic bias between two assessments conducted on the same day, and unpaired *t*-test and Mann–Whitney *U*-test were used to compare differences between the two raters’ results. Fisher’s exact test was used to compare categorical data.

Intraclass correlation coefficients were calculated to describe reliability. The ICC_1.1_ model was based on the methodology of the study; a multicenter study in which assessments were conducted at three centers, in different home-based settings, and all raters evaluated different subsets of patients [24,25].

The ICC for intra-rater reliability was calculated from the best performance at trial 1 and trial 2 registered on T1. The ICC for inter-rater reliability was calculated from the highest value registered on T1 and T2, according to recommendations [7]. The ICC_1.1_ is a fixed model addressing both systematic and random errors. ICC values were classified as follows: 0–0.5 weak; ≥0.5–0.75 moderate; ≥0.75–0.9 good; and ≥0.9 excellent reliability [24]. Agreement between results was calculated as SEM and the SEM_95_ (SEM_95_) to assess the typical error in a single measurement of repeated measurements [25,26]. Agreement was considered acceptable if below the established MCID/MID. The smallest real difference (SRD) and the SRD_95_ were calculated by the equations SRD: √2 × SEM and SRD_95_ and 1.96 × √2 × SEM, respectively, to express the variation with 95% certainty for each individual representing the smallest change to be detected beyond the measurement error [26,27,28]. SEM and SRD_95_% are presented in actual units, and expressed as a percentage of the mean of the two test sessions (grand mean), to facilitate comparisons. Bland–Altman plots were used to visualize potential systematic bias around the zero line as well as heteroscedasticity. Identification of the mean difference with 95% CI and 95% LOA were included in the plots [25]. The significance level was set as *p* < 0.05 for all analyses. Statistical analysis was performed using SPSS version 29.0.1.0 (IBM Corporation, Armonk, NY, USA).

## 3. Results

### 3.1. Participants Characteristics

A total of 50 patients participated in the intra- and inter-rater reproducibility study; female (n = 29), mean age was 71.3 (SD ± 7.7) years, FEV_1_ mean was 37.4 (SD ± 14.1), and CAT mean value was 19.4 (SD ± 6.5) points. Participants and non-participants characteristics are presented in Table 2. The two groups did not differ significantly at the baseline measurement.

### 3.2. Intra-Rater Reproducibility for 1MSTS

Measurement for intra-rater reliability achieved an excellent value of 0.95 (ICC_I.I_), agreement of 1.7 repetitions (SEM), and 2.4 repetitions (SRD) (Table 3). No systematic bias for 1MSTS was revealed in the Bland–Altman plot with 95% LOA (Figure 2). In end self-perceived dyspnea, an increase (*p* < 0.05) was found, whereas no significant increase was observed in end heart rate and dyspnea score (Table 3). Four participants were not able to perform the 1MSTS. The analysis revealed no systematic bias or relevant learning effect above the established MID when the tests were conducted with a rest of minimum 30 min between test 1 and test 2.

### 3.3. Inter-Rater Reproducibility

The results of inter-rater reliability and agreement of the 1MSTS, HGS, and SPPB tests are presented in Table 4 (T1 vs. T2).

For the 1MSTS test an excellent inter-rater reliability ICC_1.1_ of 0.91 was found, the agreement was 2.52 repetitions (SEM), and 3.5 repetitions for SRD. The Bland–Altman plot with 95% LOA revealed no heteroscedasticity and no systematic bias (Figure 2).

For the HGS test a good inter-rater reliability ICC_1.1_ of 0.84 was found; the agreement was 4.2 kg (SEM) and 5.9 kg. (SRD). The Bland–Altman plot with 95% LOA revealed no heteroscedasticity and no systematic bias (Figure 2).

For the SPPB test (total score) a good inter-rater reliability ICC_1.1_ of 0.86 was found; the agreement was 0.9 points (SEM) and 1.3 points (SRD). Sub-scores of SPPB test (balance, 3MGS, and 5STS) were explored and ICC, SEM, and SRD calculated (Table 4). Bland–Altman plot with 95% LOA revealed no heteroscedasticity and no systematic bias (Figure 2). Eight participants were not able to complete the required sit-to-stand transitions in the 5STS, i.e., the four patients that could not perform 1MSTS and additionally four other patients who could not perform 5STS. No significant differences were found in any test from T1 to T2. No significant difference was found in end saturation, heart rate, or self-perceived dyspnea (Table 4).

## 4. Discussion

This is one of the first studies to examine full reproducibility regarding measures of 1MSTS, HGS, and SPPB in patients with moderate to very severe COPD in a home-based setting. Our results demonstrate good to excellent intra-rater and inter-rater reliability for the three tests, and acceptable agreements below the established MCID/MID. Our findings support the use of these tests across different raters and settings, including home-based environments.

### 4.1. MSTS Intra-Rater and Inter-Rater Reliability

Based on our findings the inter-rater reliability test indicated an excellent ICC (ICC_1.1_ of 0.91) and acceptable agreement of the 1MSTS, and these psychometric parameters are not well reported within the COPD population. Nguyen et al. investigated reproducibility of 1MSTS and found an excellent intra-rater ICC of 0.95 (95% LOA: −1.5:1.4) and inter-rater ICC of 0.96 (95% LOA −1.9:1.4) compared to our findings. However, the participants in Nguyen’s study were slightly younger (mean age of 66.9 years), had lower disease severity (mean FEV_1_% of 57.4), and were predominately male (86%) [29], suggesting a population with a higher functional capacity which might influence reproducibility by performing tests more steadily with reduced variability. Conversely, patients with lower functional capacity could fluctuate more in health status, thus attributing to the differences in the reproducibility results.

Our SEM values of 2.5 repetitions are below earlier reported MCID/MID values (three repetitions) [13,14], which indicate a good precision of our measurements, and it suggests that repeated measurement in the same patient would vary ±2.5 repetitions on average. The differences between our COPD population and the studies above include a slightly higher age and a lower FEV_1_ pred.% in our participants who were, in addition, unable to attend center-based PR, thus being a more vulnerable group. When investigating the smallest real difference (SRD), we found a value of 3.5 repetitions and other studies have reported similar findings, but in other patient populations (systemic sclerosis (ICC 0.97; SEM: 1.1; SRD: 2.9) [30], interstitial lung disease (ICC 0.94; SEM: 1.0; SRD: 2.9) [31], and coronary heart disease (ICC 0.98; SEM: 1.4; SRD: 3.9) [32].

To our knowledge it has not previously been recommended to perform 1MSTS twice on the same day for patients with COPD; however, our results support this approach in patients with moderate to very severe COPD. The excellent reliability suggests that one test may be useful and sufficient for a clinical assessment, for instance due to time-constraints.

### 4.2. Handgrip Strength

Our results indicated good inter-rater reliability (ICC_1.1_ of 0.84, agreement was 4.2 kg (SEM) and 5.9 kg (SRD)). No well-established and consensus-based MCID/MID exists for handgrip strength specifically for the COPD population, as few studies have investigated this. However, a systematic review suggested changes from 5.0 to 6.5 kg may be meaningful, but the included studies had high heterogeneity, limiting directly generalizability to patients with COPD [33].

Karagiannis et al. investigated HGS in COPD and reported a higher intra-rater reliability of ICC 0.99, a lower SEM of 0.59 Kgf, and SRD of 1.64 Kgf as compared to our findings. The possible differences could be attributed to the study design; our study had a larger sample size (50 vs. 19), higher disease severity (FEV_1_ pred% 37.4% vs. 42.7%), more equal gender distribution (M/F: 29/21 vs. 18/1), and a slightly higher age (71.3 vs. 66.9 yrs.) [18]. Another difference between the studies was the instrument used in measuring HGS. In our study we used a handheld dynamometer (hydraulic), and the raters recorded the scoring manually, whereas in the study by Karagiannis et al., the data collection was conducted by a blinded rater and the scores were indicated on a digital screen (rounded up to the nearest 100 g) [18]. Hydraulic dynamometers may have more reader errors, which can lead to slightly over- or underestimation.

Medina-Mirapeix et al. investigated inter-rater reliability and reported an ICC 0.97 (95% CI: 0.93–0.98) and a SEM value of 1.3 kg; however, it must be noted that a different handheld dynameter model was used (ours Jamar vs. digital Nicholas Manual Muscle Tester, model 01160; Lafayette Instrument) which could attribute to different values as well as the investigated population (their participants were younger: 67 yrs., all male, and a smaller sample size (n = 30) [19].

### 4.3. SPPB—Inter-Rater Reliability

Based on our findings the SPPB demonstrated a good inter-rater reliability (ICC_1.1_ of 0.86) and a relatively low SEM (0.86) for total score. Compared to the study conducted by Medina-Mirapeix et al. our values were slightly lower for ICC (0.86 vs. 0.92) and higher for SEM (0.86 vs. 0.55 points), whereas the subscales’ ICC for balance, gait, and chair test were either divergent or aligned (balance 0.33 vs. our 0.66; gait 0.75 vs. our 0.79; and 5STS 0.84 vs. our 0.84) [19]. The population in the two studies differed on gender distribution (their: only males), respiratory symptoms (CAT score 12.7 vs. our score 19.4), and disease severity (their FEV_1_ pred % 53.8 vs. our 43.5). The differences could account for the variability in results. Furthermore, their assessments were conducted by two raters whereas ours had several more.

Our findings of a SEM of 0.86 points within the lower range from the earlier estimated MID as a total score ranging from 0.83 to 0.96 points have been proposed [34]. A study in chronic kidney disease has reported a SEM of 0.72 points and MDC90 (90% confidence interval) of 1.7 points [35]. These values provide a reference for what might be expected in similar populations with chronic conditions, but direct extrapolation to COPD should be done cautiously.

### 4.4. Strengths

The study has followed the GRAAS guidelines including a sample size of 50 patients and reporting all the relevant intra-rater and inter-rater reproducibility domains. The standardization methods were rigorous, i.e., using same chair height, same handheld dynamometer, tests performed at the same time of day, same rest between tests, standardized instruction, same rigid order of tests, and calibrated blinded raters. No patients reported exacerbation during the assessment period, and therefore, clinical stability was assumed.

To limit the influence of fatigue and dyspnea, all raters ensured that every patient had normalized oxygen saturation, heart rate, and perceived dyspnea before continuing the test.

### 4.5. Limitations

We cannot rule out the influence of fatigue as the test battery was extensive and the results might have been different if the assessments had only focused on the physical tests, thus reducing possible cognitive fatigue from the questionnaires used in the original study. However, the repeated measurements were all performed in the same order and measurement errors with potential fatigue were assumed stable. It was not possible to blind the raters or patients to the results from trial 1 and trial 2 performed same day, which increases the risk of recall bias. Finally, variables such as anxiety, depression, and respiratory symptoms were not adjusted for.

## 5. Conclusions

In conclusion, the intra- and inter-rater reliability of the 1MSTS, handgrip strength, and SPPB were good to excellent for patients with moderate to very severe COPD. These findings highlight the utility and feasibility of these tests as reliable outcome measures in clinical practice and their applicability in home-based settings.

## Figures and Tables

**Figure 1 jcm-14-06755-f001:**
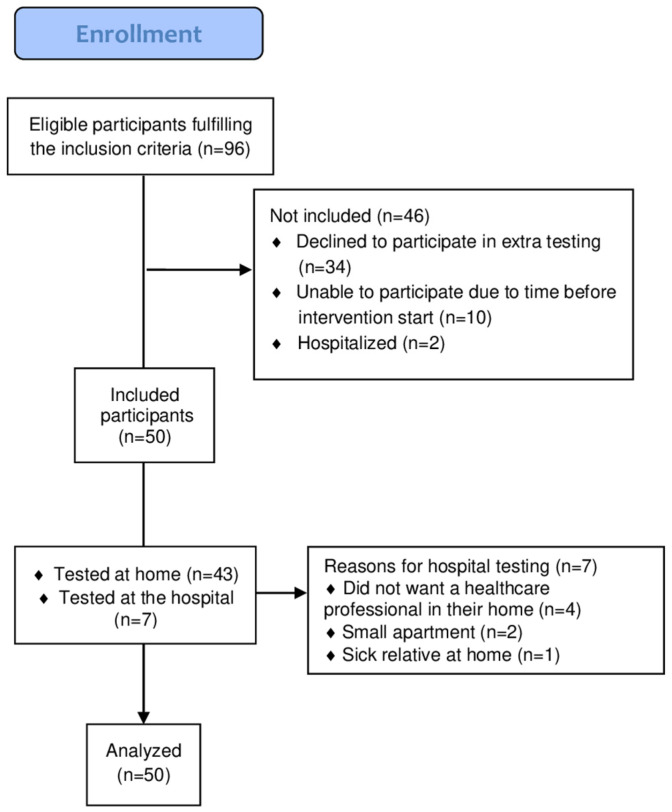
Recruitment flowchart.

**Figure 2 jcm-14-06755-f002:**
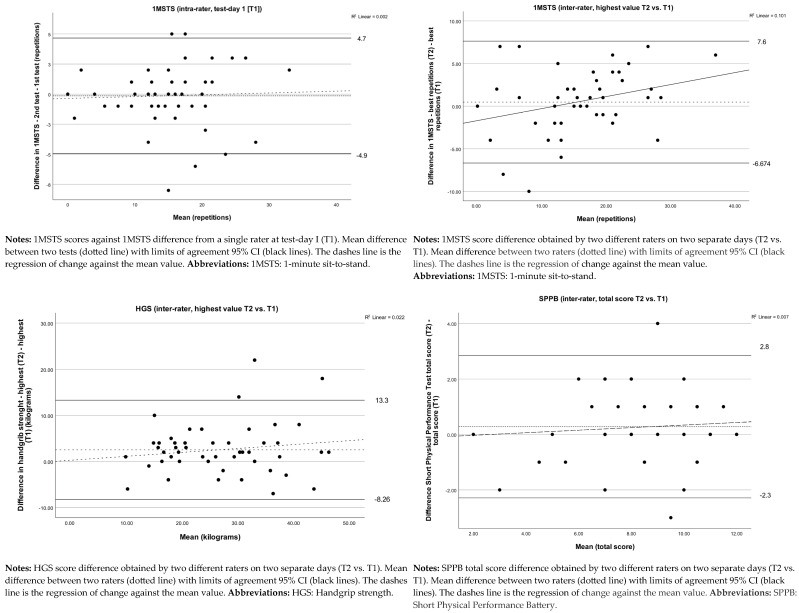
Bland-Altman plots of the 1MSTS, HGS and SPPB.

**Table 1 jcm-14-06755-t001:** Full assessment and progression of tests.

1	Participant history—only obtained at baseline test. Introduction to the tests, performed while seated. Measurement seated: resting blood pressure, resting heart rate, resting SpO_2_, resting dyspnea (10 min).
2	Introduction and performing: 1MSTS. Measurement seated: end-heart rate, end-SpO_2_, end-dyspnea (10 min).
3	Introduction and performing: HGS (10 min).
4	Seated rest while completing questionnaires (CAT, HADS, EQ5D-3L) (15 min).
5	Seated: resting blood pressure, resting heart rate, resting SpO_2_, resting dyspnea (5 min).
6	Introduction and performing: 1MSTS. Measurement seated: end-heart rate, end-SpO_2_, end-dyspnea (5 min).
7	Seated rest while completing questionnaires (PSQI, MFI, BPI) (15 min).
8	Introduction and performing: SPPB (10 min).
9	Assessment session completed: total time approximately 75–90 min.

**Notes**: The table provides an overview of full assessment. In the present article, only physical tests are reported (1MSTS, HGS, and SPPB). Only 1MSTS was performed twice, unlike HGS and SPPB due to the length and time of the full assessment. **Abbreviations**: SpO_2_: arterial oxygen saturation as measured by pulse oximetry (%); HR: heart rate (beasts pr. minute); dyspnea: perceived dyspnea (Borg cr-10 scale); 1MSTS: 1 min sit-to-stand; HGS: handgrip strength; CAT: COPD assessment test; HADS: hospital anxiety and depression scale; EQ-5D: EuroQol 5-dimensions 3-Likert utility score and VAS score; PSQI: Pittsburgh sleep quality questionnaire; MFI: multidimensional fatigue inventory; BPI: brief pain inventory; SPPB: short physical performance battery (score).

**Table 2 jcm-14-06755-t002:** Characteristics of included and non-included patients in study.

Variables	Patients Included in Reproducibility Study	Patients NOT Included in Reproducibility Study
Sex, male/female (n)	21/29 (50)	15/31 (46)
Age, years, mean (SD)	71.3 (±7.7)	70.0 (±8.9)
Body mass index, kg/m^2^, mean (SD)	25.9 (±5.9)	25.4 (±6.1)
FEV_1_, % predicted, mean (SD)	37.4 (±14.1)	34.5 (±12.0)
FEV_1_/FVC, mean (SD)	43.5 (±10.8)	41.9 (±10.3)
GOLD I/II/III/IV, %	0/22/40/38	0/15/46/39
A/B/C/D, %	0/50/2/48	0/28/11/61
MRC dyspnea scale, median (range)	4.0 (2–5)	4.0 (2–5)
CAT symptoms, mean (SD)	19.4 (±6.5)	21.3 (±5.8)
BODS index points, mean (SD)	5.2 (±2.1)	5.8 (±1.7)
Charlson index, 0/1/2 ≥ 3, %	2/30/42/26	0/54/17/29
LTOT, n (%)	6 (12)	9 (20)
Walking aid, stick/walker n (%)	23 (46)	23 (50)
Walking aid during SPPB gait test, n (%)	5 (22)	4 (17)
Highest 1MSTS, (SD)	15.2 (7.6)	12.7 (8.6)
Highest HGS, (SD)	26.7 (10.1)	24.8 (8.9)
Highest SPPB point, (SD)	8.6 (2.5)	8.3 (2.8)

**Notes:** Data are presented as mean ± SD, median (range), or percentages in non-normally distributed variables. **Abbreviations:** FEV_1_: forced expiratory volume in one second; FVC: forced vital capacity; FEV_1_/FVC: ratio between FEV_1_ and FVC; MRC: medical research council dyspnea scale; CAT: COPD Assessment test, BODS index: body-mass, airflow obstruction, dyspnea, and severity, LTOT: long-term oxygen therapy, 1MSTS; 1 min sit-to-stand, HGS: handgrip strength, SPPB: short physical performance battery.

**Table 3 jcm-14-06755-t003:** Intra-rater reproducibility test-day I (T1).

Variables	Test One	Test Two	Difference	Floor Effect (n) (%)	ICC_I.I_ (LL_95_)	SEM (SEM%)	SEM_95_ (SEM_95_%)	SRD (SRD%)	SRD_95_ (SRD_95_%)
1MSTS	14.4 (± 7.5)	14.3 (±7.6)	−0.14 [−0.84; 0.56]	4 (8)	0.95 (0.91)	1.7 (12)	3.3 (23)	2.4 (17)	4.7 (33)
End S_p_O_2_, (after STS)	88.9 (±13.8)	89.7 (±14.0)	0.77 [0.06; 1.48] *	NA	0.98 (0.97)	1.9 (2)	3.7 (4)	2.7 (3)	5.3 (6)
End HR (after STS)	98.6 (±22.1)	99.5 (±22.4)	0.88 [−1.72; 3.39]	NA	0.92 (0.87)	6.1 (6)	12.0 (12)	8.6 (8)	16.7 (17)
End dyspnea score (after STS)	4.9 (±2.0)	5.2 (±2.0)	0.27 [−0.07; 0.61]	NA	0.83 (0.72)	0.8 (16)	1.5 (30)	1.1 (22)	2.2 (44)

**Notes:** Data for test one and test two are presented as mean ± SD, and differences as delta [SE 95%]. Significant differences between tests are denoted as * *p* ˂ 0.05. **Abbreviations:** 1MSTS: 1 min sit-to-stand (repetitions); S_p_O_2_: arterial oxygen saturation as measured by pulse oximetry (%); HR: heart rate (beasts pr. minute); dyspnea: perceived dyspnea (Borg cr-10 scale); ICC I.I: intraclass correlation coefficient model I.I; LL95: lower 95% confidence limit; SEM: standard error of measurement; SEM%: standard error of measurement expressed as a percentage of the mean; SEM_95_%: standard error of measurement expressed at the 95% confidence interval; SRD: smallest real difference; SRD%: smallest real difference as a percentage of the mean; SRD_95_%: smallest real difference at the 95% confidence level.

**Table 4 jcm-14-06755-t004:** Inter-rater reproducibility (highest value T1 vs. highest value T2).

Variables	Rater T1	Rater T2	Difference	Floor (n) (%)	ICC_I.I_ (LL_95_)	SEM (SEM%)	SEM_95_ (SEM_95_%)	SRD (SRD%)	SRD_95_% (SRD_95_%)
1MSTS	15.2 (±7.8)	15.7 (±8.9)	0.48 [−2.87; 3.83]	4 (8)	0.91 (0.84)	2.52 (16)	4.9 (31)	3.5 (22)	6.9 (44)
End S_p_O_2_, (after STS)	89.4 (±13.8)	87.3 (±19.2)	−2.1 [−8.89; 4.64]	NA	0.66 (0.46)	9.6 (11)	18.8 (21)	13.6 (15)	26.6 (30)
End HR (after STS)	99.5 (±22.4)	98.6 (±22.5)	−0.39 [−10.70; 9.90]	NA	0.63 (0.43)	10.1 (10)	19.8 (20)	14.3 (14)	12.3 (12)
End dyspnea score (after STS)	5.1 (±2.0)	4.9 (±2.5)	−0.28 [−1.22; 0.65]	NA	0.63 (0.42)	1.4 (28)	2.7 (54)	1.9 (38)	3.9 (78)
Highest HGS, dominant hand, kg.	26.7 (±10.1)	29.3 (±11.0)	2.5 [−1.67; 6.71]	NA	0.84 (0.74)	4.2 (15)	8.2 (29)	5.9 (21)	11.6 (41)
SPPB, total score	8.6 (±2.5)	8.9 (±2.6)	0.28 [−0.72; 1.28]	NA	0.86 (0.77)	0.9 (10)	1.8 (20)	1.3 (15)	2.5 (29)
SPPB balance, score	3.48 (±0.9)	3.46 (±0.9)	0.02 [−0.39; 0.35]	NA	0.66 (0.47)	0.5 (14)	0.9 (26)	0.7 (20)	1.4 (40)
SPPB 3-m gait, score	2.6 (±0.9)	2.8 (±0.9)	0.20 [−0.16; 0.56]	NA	0.79 (0.65)	0.4 (15)	0.8 (30)	0.6 (23)	1.2 (45)
SPPB 5-times-STS, score	2.6 (±1.5)	2.7 (±1.5)	0.10 [−0.49; 0.69]	8 (16)	0.84 (0.74)	0.6 (23)	1.1 (41)	0.8 (30)	1.5 (56)

**Notes:** Test one (T1), test two (T2), and differences are presented as mean ± SD or delta difference [SE 95%]. **Abbreviations:** 1MSTS: 1 min sit-to-stand (repetitions); HGS: handgrip strength (kilograms); SPPB: short physical performance battery (score); S_p_O_2_: arterial oxygen saturation as measured by pulse oximetry (%); HR: heart rate (beasts pr. minute); dyspnea: perceived dyspnea (Borg cr-10 scale); ICC I.I: intraclass correlation coefficient model I.I; LL95: lower 95% confidence limit; SEM: standard error of measurement; SEM%: standard error of measurement expressed as a percentage of the mean; SEM_95_%: standard error of measurement expressed at the 95% confidence interval; SRD: smallest real difference; SRD%: smallest real difference as a percentage of the mean; SRD_95_%: smallest real difference at the 95% confidence level.

## Data Availability

Data access in Denmark are under strict juristic data protection law as imposed by the Danish Ministry of Justice. Any possible access or sharing demands a part-application to (1) Danish Data Protection Agency (email dt@datatilsynet.dk), (2) Ethics Committee of the Capital Region (email vek@regionh.dk), and (3) National Health Data Authorities (email kontakt@sundhedsdata.dk). Only if the applications are approved will the data be considered available for sharing.

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
