# Peer review of "Intra- and Inter-Rater Reproducibility of Measures of Physical Performance in Patients with COPD"

_jcm, 2025, doi:10.3390/jcm14196755_

Round 1
Reviewer 1 Report
Comments and Suggestions for Authors
Thank you for the opportunity to review this manuscript. In this study, the authors aim to investigate the intra- and inter-rater reproducibility of three field tests, that are well-know and widely used in clinical practice. The authors specify that this topic is developed in particular in a "home-based" context. Overall, the investigated topic is relevant with the development of home-based interventions for COPD patients since two decades and the study offers valuable insights that could support the use of these specific field tests in home context. The article is well-written and provides clear information for an easy and comprehensive read. The methods are relevant for the main objective of the study, with a test-retest approach. However, several issues must be addressed. Please find my comments below, part by part :
Abstract
- Please indicate why these field tests have been chosen.
Introduction
- L.62-64 and L.83-85 : The authors reported two previous studies about the intra-class coefficients for 1MSTS in COPD population. If ICC values have been already reported in the literature, the authors may have to state how innovative is the current work. This may also concern HGS and SPBB. A deep read of the introduction may suggest that the "home-based" context could be the original approach of this study. If so, the authors should indicate factors associated with home-based context that may have an influence on reproducibility of the mentioned field tests (L.83-85).
- If applicable, please state if there is any previous data for inter-rater reliability for 1MSTS and for intra-rater reliability for SPPB or not.
Materials and methods
- L.106-108 : please provide some extra information about the choice of 50 patients’ sample size.
- Figure 1 (L.100-117) : the authors want to position the study in a "home-based" context and mention in the introduction part that research on home-based assessment is limited. Finally, the objective of the study clearly mentions that the evaluation of the intra-/inter-rater reliability will be made "in a home-based setting". It’s therefore surprising to observe that 7 patients of the 50 patients included (approximately 15% of the tested population) had been tested at the hospital. This point has to be discussed by the authors as this contradicts what is clearly announced in the first part of the manuscript. Why did the authors offer the opportunity to perform the tests in hospital (L.128-130)? Could this influence the primary outcome as the conditions at the hospital may be clearly different from home-based setting.
- The numerous inclusion criteria could call into question the generalisability and the external validity of the reported results. In this way it could be hard to use the chosen tests for patients with other disease severity status. This point could therefore be included in the Limitations part.
- L.211-212 : please make sure that the sentence is well written as the sentence is separated in two different parts in the reviewing version.
- L.216-217 : Could the authors provide a reference for the classification of ICC values?
- After reviewing the manuscript, a global question about methodological choices emerges : why was intra-rater reproducibility only tested for 1MSTS and not HGS and SPPB? The design of the general RCT may allows to do a complete reproducibility study for the 3 field tests and the main objective of the study doesn’t state about this aspect.
Results
- Table 3 : What does the variable " Floor" mean exactly? Unless I’m mistaken, I don’t read any explanation about this variable in the Methods part.
- Bland-Altman plots are really interesting, in order to observe the distribution of points in intra- or inter-rater procedure. Globally, these figures confirm the good (or excellent) reproducibility of the 3 field tests, but with a few outliers that go beyond agreement limits. However, the regression line of change for inter-rater reproducibility of 1MSTS displays an increasing difference between T2 and T1 when the mean performance increases. This appears to be an interesting secondary point of the study.
Discussion
- L.303 : Once again, the authors state that the ongoing study is based on a home-based setting. From my point of view, this could be controversial as 15% of the patients performed their tests in hospital-based setting and moreover, the authors offered the opportunity to the patients to be evaluated at hospital.
- L.311-316 : The authors referred to a previous study of Nguyen et al. for comparing their results to the literature. The authors reported older patients, with more severe COPD disease and sex differences for explaining difference between intra- and inter-rater values. This aspect must be deepen, by explaining in detail how these mentioned parameters (age, disease severity or sex repartition) could lead to an improved or decreased reproducibility for 1MSTS. A simple vision of this point will be to say that patients with higher functional capacity (as stated by the authors) will reach higher performance and inversely. It seems important to explain whether higher or lower functional capacity could influence reproducibility.
- L.327-330 : this paragraph is of great importance as it points out an important take-home message of the ongoing study. The authors might consider to report this key point in the conclusion part.
Author Response
Dear Reviewer
Thank you for your time reviewing this manuscript. Your comments are of great aprreciation. Please find answers to comments below.
Kind regards.
Christina
Comment 1 (abstract)
Please indicate why these fields have been chosen.
Answer 1
Thank you for that comment. The reasoning for choosing the 1MSTS is, that the test is recommended as a validated proxy for 6MWT, as an expression for functional capacity. The SPPB test battery is recommended by the ERS/ATS to screen for frailty and measure changes over time. We do expect frailty to be more present in this selected population and a secondary research question that lack evidence. The HGS, like the SPPB, is also recommended for frail patients, but evidence lack in a population with more disease severity.
We unfortunately have no more space in the abstract to indicate the reasoning for these tests, but hope it is clearly stated in the background.
Comment 2 (introduction)
L 62-64 and L 83-85: The authors have reported previous studies about the intra-class coefficient for 1MSTS in COPD population. If ICC values have been already reported in the literature, the authors may have to state how innovative is the current work. This may also concern HGS and SPBB. A deep read of the introduction may suggest that the “home-based” context would be the original approach of this study. If so, the authors should indicate factors associated with home-based context that may have an influence on reproducibility of the mentioned field tests (L 83-85).
Answer 2
Thank you for that comment. I have added a bit more information. Please see L 55-57, L 66-67 and L85-90 in the uploaded revised version.
Comment 3
If applicable, please state if there is any previous data for inter-rater reliability for 1MSTS and for intra-rater reliability for SPPB or not.
Answer 3
Thank you for that comment. I have added a bit more information, please see L66-67, L85-91.
Materials and methods
Comment 4
L 106-108: please provide some extra information about the choice of 50 patients’ sample size.
Answer 4:
Thank you for that comment. I have added some extra information of the chose sample size and hope it clarifies the chosen number, please see L114-115.
Comment 5
Figure 1 (L 100-117): the authors want to position the study in a “home-based” context and mention in the introduction part that research on home-based assessment is limited. Finally, the objective of the study clearly mentions that the evaluation of the intra/inter-rater reliability will be made “in a home-based setting”. It’s therefore surprising to observe that 7 patients of the 50 patients included (approximately 15% of the tested population) had been tested at the hospital. This point has to be discussed by the authors as this contradicts what is clearly announced in the first part of the manuscript. Why did the authors offer the opportunity to perform the tests in the hospital (L 128-130)? Could this influence the primary outcome as the conditions at the hospital may be clearly different from home-based setting.
Answer 5
Thank you for that comment. The participants were given the choice of “test setting” to accommodate their individual preferences. Our primary target was to assess the participants at home, however if the participant did not want a HP in their home for reasons such as a sick relative etc., we did not exclude them from taking part in the study. I have updated the CONSORT flow diagram with these information’s, please see page 5.
Comment 6
The numerous inclusion criteria could call into question the generalizability and the external validity of the reported results. In this way it could be hard to use the chosen tests for patients with other disease severity status. This point could therefore be included in the limitations part.
Answer 6
Thank you for that comment. It is a very fair point, and we acknowledge your thoughts about the generalizability and external validity. We do not have plausible reasons to believe that persons with lower disease severity should be more unstable/higher variability from week to week. However, we would from an empirical clinical perspective assume that the persons with worse disease severity (our population) would exhibit more variability from week to week as regards to function and self-reported symptoms. In the ideal world it would be very relevant to confirm the findings in a lower disease population, however in Denmark these are followed by the general practitioner.
Comment 7
L 211-212: please make sure that the sentence is well written as the sentence is separated in two different parts in the reviewing version.
Answer 7
Thank you for spotting that. It has now been corrected, please see L243-244.
Comment 8
L 216-217: Could the authors provide a reference for the classification of ICC values?
Answer 8
Thank for your comment. I have provided a reference for the ICC values in the manuscript, please see L249 (reference to Koo et al).
Comment 9
After reviewing the manuscript, a global question about methodological choices emerges: why was intra-rater reproducibility only tested for 1MSTS and not HGS and SPPB? The design of the general RCT may allow to do a complete reproducibility study for the 3 fields tests and the main objective of the study doesn’t state about this aspect.
Answer 9
Thank you for that comment. We chose to only test intra-rater reproducibility on 1MSTS due to the length of the full test assessment. The time to complete were app. 1,5 hours – after which participants were mentally and physically tired. We therefore chose that 1MSTS would be the most valuable test to investigate for intra-rater reproducibility.
Results
Comment 10
Table 3: What the variable “Floor” mean exactly? Unless I’m mistaken, I don’t read any explanation about this variable in the Methods part.
Answer 10
Thank you for that comment. We do not explicitly address the variable floor in our manuscript but have chosen to keep the values in our tables to provide transparency, as four patients were not to perform the 1MSTS test.
Comment 11
Bland-Altman plots are really interesting, in order to observe the distribution of points in intra- or inter-rater procedure. Globally, these figures confirm the good (or excellent) reproducibility of the 3 fields test, but with a few outliers that goes beyond the agreement limits. However, the regression line of change for inter-rater reproducibility of 1MSTS displays an increasing difference between T2 and T1 when the mean performance increases. This appears to be an interesting secondary point of the study.
Answer 11
Thank you for that comment. We fully agree that the increasing difference between T2 and T1 is an interesting point. However, our analysis revealed no systematic bias or learning effect. The observed variability in measurements may instead be related to factors such as comorbidity symptoms or differences in participant motivation.
Discussion
Comment 12
L 303: Once again, the authors state that the ongoing study is based on a home-based setting to the literature. From my point of view, this could be controversial as 15% of the patients performed their tests in hospital-based setting and moreover, the authors offered the opportunity to the patients to be evaluated at hospital.
Answer 12
Thank you for that comment. The aim was to investigate reproducibility in a home-setting, however seven patients chose to be tested at the hospital with the primary reason being privacy (the consort diagram have been updated with information on their decline to be tested at the hospital). Mixing different test settings can provide problems such as environmental variability, however the use of model ICC1,1, addresses variability from differing environments and our stringent use of a standardized protocol should limit variabilities.
Comment 13
L 311-316: The authors referred to a previous study of Nguyen et al for comparing their results to the literature. The authors reported older patients, with more severe COPD disease and sex differences for explaining the difference between intra- and inter-rater values. This aspect must be deepen, by explaining in detail how these mentioned parameters (age, disease severity or sex repartition) could lead to an improved or decreased reproducibility for 1MSTS. A simple vision of this point will be to say that patients with higher functional capacity (as stated by the authors) will reach higher performance and inversely. It seems important to explain whether higher or lower functional capacity could influence reproducibility.
Answer 13
Thank you for that comment. I have added more information to address this important point, please see L348-351 and L356-357.
Comment 14
L 327-330: this paragraph is of great importance as it points out an important take-home message of the ongoing study. The authors might consider to report this key point in the conclusion part.
Answer 14
Thank you for that comment. We would like to keep the conclusion short and will therefore keep the original conclusion.
Reviewer 2 Report
Comments and Suggestions for Authors
I read this paper about home based assessment of physical performance in COPD with great interest. The paper is well written and wel-designed and provides useful data.
However, I have some major concerns which authors must address:
-The aim of the study is stated, but no clear hypothesis is written. Please modify.
-The use of ICC1,1 should be justified more clearly. Why not choosing ICC2,1 or ICC3,1? Please comment on that.
-The supplementary file is identical to the manuscript file!
- I found some English errors throughout the paper. Please have a deep language revision.
Comments on the Quality of English Languagemoderate revision required
Author Response
Dear Reviewer
Thank you for taking the time to review our manuscript, it is greatly appreciated.
I have written answers to your comments below.
Kind regards,
Christina
Comment 1
The aim of the study is stated, but no clear hypothesis is written. Please modify.
Answer 1
Thank you for your comment concerning a hypothesis. Our main aim of the study was to investigate reproducibility and not test a novel hypothesis. The lack of hypothesis does not imply a lack of scientific rigor as we have followed the recommended guidelines. In the process of including a hypothesis in our manuscript I have with interest reviewed the referenced articles from the background and none of them generate a hypothesis. If it is critical to include in our manuscript a suggestion would be: We hypothesized that the tests would achieve acceptable agreement.
Comment 2
The use of ICC1,1, should be justified more clearly. Why not choosing ICC2,1 or ICC3,1? Please comment on that.
Answer 2
Thank you for your comment on our choice of model ICC1,1. The reasoning for choosing ICC1,1 was because the assessments were conducted across three centers as well as in patients’ homes, with different raters evaluating different subsets of patients.
The ICC2,1 and ICC3,1 models were not selected, as these require the same raters to assess all subjects, which was not feasible in this multicenter study involving multiple raters with varying assignments. Additionally, ICC2,1 assumes raters as random effects generalizable to a larger population, and ICC3,1 treats raters as fixed effects specific to the study, neither of which aligned with the study’s design where rater overlap was incomplete.
I have made some adjustment in the manuscript to be more transparent about the reasoning for our choice, please see L242-244.
Comment 3:
The supplementary file is identical to the manuscript file!
Answer 3:
Thank you for your comment. That was an error when uploading the original manuscript. No supplementary files are a part of the manuscript.
Comment 4:
I have found English errors throughout the paper. Please have a deep language revision.
Answer 4:
Thank you for your comment. We have read the manuscript and revised the English errors.
Round 2
Reviewer 2 Report
Comments and Suggestions for Authors
Authors correctly replied to my comments. Ok to accept for me now.